# Long-Term Adherence to a Mediterranean Diet 1-Year after Completion of the MedLey Study

**DOI:** 10.3390/nu14153098

**Published:** 2022-07-28

**Authors:** Karen J. Murphy, Kathryn A. Dyer, Belinda Hyde, Courtney R. Davis, Ella L. Bracci, Richard J. Woodman, Jonathan M. Hodgson

**Affiliations:** 1Alliance for Research in Exercise, Nutrition and Activity, University of South Australia, GPO Box 2471, Adelaide, SA 5001, Australia; courtney.davis@unisa.edu.au (C.R.D.); ella.bracci@mymail.unisa.edu.au (E.L.B.); 2Clinical and Health Sciences, University of South Australia, GPO Box 2471, Adelaide, SA 5001, Australia; kate.dyer@unisa.edu.au; 3Allied Health and Human Performance, University of South Australia, GPO Box 2471, Adelaide, SA 5001, Australia; belinda.hyde@unisa.edu.au; 4Flinders Centre for Epidemiology and Biostatistics, Flinders University, GPO Box 2100, Adelaide, SA 5001, Australia; richard.woodman@flinders.edu.au; 5Nutrition and Health Innovation Research Institute, School of Medical and Health Sciences, Edith Cowan University, 35 Stirling Highway, Perth, WA 6000, Australia; jonathan.hodgson@ecu.edu.au; 6Medical School, University of Western Australia, 35 Stirling Highway, Perth, WA 6000, Australia

**Keywords:** Mediterranean diet, MedLey study, Australians, dietary intervention trial

## Abstract

Mediterranean populations enjoy the health benefits of a Mediterranean diet (MedDiet), but is it feasible to implement such a pattern beyond the Mediterranean region? The MedLey trial, a 6-month MedDiet intervention vs habitual diet in older Australians, demonstrated that the participants could maintain high adherence to a MedDiet for 6 months. The MedDiet resulted in improved systolic blood pressure (BP), endothelial dilatation, oxidative stress, and plasma triglycerides in comparison with the habitual diet. We sought to determine if 12 months after finishing the MedLey study, the participants maintained their adherence to the MedDiet principles and whether the reduction in the cardiovascular disease (CVD) risk factors that were seen in the trial were sustained. Participants completed a food frequency questionnaire, and a 15-point MedDiet adherence score (MDAS; greater score = greater adherence) was calculated. Home BP was measured over 6 days, BMI was assessed, and fasting plasma triglycerides were measured. The data were analysed using intention-to-treat linear mixed effects models with a group × time interaction term, comparing data at baseline, 2, 4, and 18 months (12 months post-trial). At 18 months (12 months after finishing the MedLey study), the MedDiet group had a MDAS of 7.9 ± 0.3, compared to 9.6 ± 0.2 at 4 months (*p* < 0.0001), and 6.7 ± 0.2 (*p* < 0.0001), at baseline. The MDAS in the HabDiet group remained unchanged over the 18-month period (18 months 6.9 ± 0.3, 4 months 6.9 ± 0.2, baseline 6.7 ± 0.2). In the MedDiet group, the consumption of olive oil, legumes, fish, and vegetables remained higher (*p* < 0.01, compared with baseline) and discretionary food consumption remained lower (*p* = 0.02) at 18 months. These data show that some MedDiet principles could be adhered to for 12 months after finishing the MedLey trial. However, improvements in cardiometabolic health markers, including BP and plasma triglycerides, were not sustained. The results indicate that further dietary support for behaviour change may be beneficial to maintaining high adherence and metabolic benefits of the MedDiet.

## 1. Introduction

The Mediterranean dietary pattern (MedDiet) is a UNESCO-heritage-listed dietary pattern shared by many populations surrounding the Mediterranean region. It is characterised by the high consumption of extra-virgin olive oil, nuts, legumes, vegetables, fruit, and whole grains, with a moderate amount of fish and poultry, low amounts of dairy foods, red and processed meats and discretionary foods, and wine consumed in moderation and with meals [1,2]. The MedDiet is also associated with knowledge, skills, and traditions around food production, harvesting and processing, a lifestyle of physical activity, communal meals with friends and family, conviviality, and frugality [2,3]. It has been promoted as one of the healthiest dietary patterns for reducing chronic disease risk and promoting healthy longevity. An umbrella review of meta-analyses of 13 observational studies and 16 randomised controlled trials was performed, exploring the link between MedDiet adherence and 37 different health outcomes in 12.8 million participants. The authors showed that higher adherence to a MedDiet was associated with reduced risk of all-cause mortality, Alzheimer’s disease/dementia, neurodegenerative diseases, cardiovascular disease, coronary heart disease, myocardial infarction, diabetes, and cancer incidence [4]. Further, better adherence to a MedDiet may lead to increased masticatory performance [5], fewer falls in community-dwelling individuals [6], better dyslipidemia and low-grade inflammation profiles in familial hypercholesterolemia [7], and a reduced risk of global cognitive decline [8]. 

Following a MedDiet and the lifestyle of Mediterranean populations may offer a challenge for Western countries that have less healthy diets and physical activity behaviours. Nutritional epidemiological data indicate that Western diets are characterised by high intakes of highly processed food, red and processed meats, added sugars, saturated fat, and alcohol with sub-optimal intakes of fruit, vegetables, and whole grains [9]. In Australia, less than 10% of Australian adults meet the recommendations for two servings of fruit and five servings of vegetables per day [10], whilst two-thirds fail to meet physical activity guidelines [11]. Further, Western countries, including Australia, New Zealand, the USA, Canada, and Western Europe, globally have a high consumption of unhealthy foods [12]. To date, several trials have explored the effect of a MedDiet on health outcomes in populations outside of the Mediterranean basin. The first MedDiet versus habitual diet study in Australia was published in 2011, a cross-over study in 27 men and women, mean age 59 years, with type 2 diabetes. The participants were provided with ~70% of pre-prepared meals based on a Greek MedDiet (%en from fat, carbohydrate, and protein: 40:44:12) for 12 weeks. The authors demonstrated a significant −0.3% reduction in HbA1c in the MedDiet group compared with the habitual diet group [13]. This was followed by the MedLey trial (Mediterranean Diet for cognitive and cardiovascular health in the elderly), an RCT with 137 men and women aged age 71 years (±4.9 years), which compared a MedDiet and habitual diet over 6 months. This trial demonstrated that with regular contact with study dietitians and a supply of MedDiet foods, the participants could easily achieve high adherence to the MedDiet, resulting in a significant reduction in blood pressure, oxidative stress, blood triglycerides, and a significant increase in endothelial dilatation [14,15,16]. Following this research, several other trials in Australia have explored a MedDiet on risk factors of CVD [17,18] and other chronic conditions, such as depression [19,20], non-alcoholic fatty liver disease [21], kidney function [22], asthma [23], secondary heart disease, and on mood as well as different populations, such as transplant patients. What is clear is that Australians can adhere to a MedDiet, but what remains unclear is whether they can maintain that level of adherence following the cessation of a research intervention. The aim of this study was to determine if, 12 months after finishing the MedLey study, the participants were still adhering to the MedDiet principles, if the habitual diet group had adopted the MedDiet pattern and whether the reduction in CVD risk factors seen in the trial was sustained.

## 2. Methods

The protocol for the MedLey Study (ACTRN12613000602729) has been described in detail elsewhere [24,25]. In brief, the MedLey study was a randomised, parallel controlled dietary intervention study conducted in Adelaide, South Australia, between 2013 and 2015. Healthy, elderly (71 ± 5, range 64–86 years) Australian men and women, screened against inclusion criteria [24,25] (*n* = 166 randomised, *n* = 152 commenced) were randomised to receive a MedDiet (MedDiet group) or to maintain their Habitual diet (HabDiet control group) for six months. The outcome measures assessed at baseline at 2 and 4 months of the intervention included a 15-point MedDiet adherence score based on weighed-food record data. The outcome measures assessed at baseline, 3 and 6 months, included blood pressure, flow-mediated dilatation, cerebral blood flow, blood lipids, glucose, insulin, *hs*-CRP, F_2_-isoprostanes, anthropometry and body composition, cognition and markers of dietary adherence including urinary metabolites, serum carotenoids and erythrocyte fatty acids. This was a dietitian-led intervention where all participants consulted the study dietitian fortnightly, received resources including recipes and meal plans, and those in the intervention arm collected study foods representing the MedDiet, including canned fish, legumes, walnut, peanut, and almond mix, Greek yoghurt, and extra virgin olive oil. 

### 2.1. Follow-Up Design

Twelve months after the completion of the MedLey study, the participants were sent a letter of invitation to participate in an optional follow-up study. The follow-up study was an unfunded initiative of the MedLey investigator team conducted by the Alliance for Research in Exercise, Nutrition and Activity, University of South Australia.

An information pack containing a letter of invitation, a food frequency questionnaire, and a follow-up questionnaire (specific to either the MedDiet or the HabDiet) were posted to the participants who completed the 6-month study. The participants were given the option to decline to participate, complete only the postal questionnaire, or to complete the questionnaire as well as attend a clinic visit. The participants who indicated they were willing to participate in the clinic visit had an appointment scheduled as close to 12 months as possible (within one week on either side) to the date that they finished the study. The participants who undertook the clinic visit attended following an overnight fast, and the following measures were taken: fasting blood sample, blood pressure, body mass (weight), cognitive testing, and for those participants of the MedLey study who had vascular data; endothelial vasodilatory function (via Flow Mediated Dilatation; FMD) and cerebral vascular responsiveness (via transcranial Doppler; TCD). Following the clinic visit, they were given a blood pressure monitor to take home with instructions to record their BP in triplicate three times daily (morning, afternoon, and evening) for 6 days.

### 2.2. Outcomes

The outcome measures were chosen based on those measures that significantly changed in the MedLey study, which included F_2_-isoprostanes, blood lipids and lipoproteins, erythrocyte fatty acids and plasma carotenoids, endothelial function, and blood pressure. As cognition showed no significant change in the MedLey study [26], the follow-up data were not analysed. F_2_-isoprostanes, erythrocyte fatty acid composition, and plasma carotenoids were not analysed due to budgetary constraints. The assessment of endothelial function using Flow-Mediated Dilatation (FMD) was discontinued during the follow-up due to equipment malfunction; therefore, the data are not reported. All of the methods were conducted according to the same procedures previously reported in the MedLey study protocol [25]. 

### 2.3. Dietary Adherence

The dietary data were collected from a validated 74-item food frequency questionnaire (FFQ), which was supplemented with additional questions regarding the intake of fish, nuts, oil, and full-sugar soft drinks [27]. MedDiet adherence was determined using a 15-point score, adapted from the 9-point MedDiet score created by Trichopoulou et al. [28] and previously described [14], using a 9-point literature-based score for comparison with other literature [29]. 

Briefly, the total food intake (as g/MJ/day) was divided into 15 food groups that define a MedDiet. The group means of these 15 food groups were calculated for MedDiet and HabDiet at baseline for use as cut-offs. Each of the participant’s intake for each food group in total g/MJ/day was compared to the group average at baseline, and 1 point was given for intakes above the group mean for vegetables, fruits, legumes, nuts, fish, breads, cereals, and olive oil and 1 point if intakes were below the group mean for sugars, eggs, dairy foods, potato, meat, and miscellaneous foods. Red wine was not calculated based on group averages but rather compared to a fixed recommendation and, therefore, calculated on the amount of 23.3 mL/MJ/day as per the MedLey protocol. A score between 0–15 was given, where 15 indicates the highest adherence to the MedDiet.

To calculate adherence to the MedDiet for an additional comparison to other published data, a 9-point literature-based score developed by Sofi and colleagues [30] was also adopted. The dietary data from the FFQ were re-grouped into the 9 food groups according to Sofi et al. [30] (vegetables, fruit, legumes, cereals, fish, meat and meat products, dairy foods, red wine, and olive oil). The sum of food per group in g/day was then scored according to the literature-based criteria for MedDiet adherence, where a score of 0, 1, or 2 was given as determined by cut-offs of g/day against serving sizes and the recommended number of servings. A score of 0, 1, or 2 was assigned across the 9 food groups associated with the Mediterranean diet. For the food groups associated with positive health outcomes (fruit, vegetables, legumes, cereals, fish, and olive oil), higher intake is awarded a higher score. For red wine, less than one serving per day is assigned a score of 1, between 1 and 2 servings per day is assigned a score of 2, and greater than 2 servings per day is assigned a score of 0. A higher total score indicates greater adherence to a MedDiet. We do acknowledge that some adherence scores negatively score adherence to certain foods, such as dairy foods; however, recent scores, such as those used in MedLey or MedDairy, positively score the consumption of dairy foods [14,31].

### 2.4. Statistical Analysis

For analyses, the two groups consisted of participants who completed the postal questionnaires only (*n* = 128 *n* = 66 MedDiet, *n* = 62 HabDiet) and participants who completed the postal questionnaires as well as attend the clinic visit (*n* = 108 *n* = 55 MedDiet, *n* = 53 HabDiet). Outliers were identified from histograms and outlier boxplots. Any outliers that were greater than three standard deviations from the mean and affected the group mean and the distribution of residuals were excluded from the analyses. In this way, *n* = 1 participant from the HabDiet group was excluded from the blood pressure data, *n* = 3 (MedDiet *n* = 2, HabDiet *n* = 1) were excluded from LDL-C data, and *n* = 1 from the MedDiet group was excluded from the dietary analysis. Where alcohol intake was assessed over the study, only those who drank alcohol were included in these analyses (zero values or non-drinkers were excluded from the analyses). Therefore, when considering how alcohol intake changed across the course of the study, we have assessed the proportion of the population who drink alcohol. 

The residuals were checked for normality, and any non-normal variables were log-transformed before analysis. The statistical results of the transformed data have been exponentiated to present meaningful values and are thus presented as a ratio of the change rather than the mean difference between time points. 

Analyses were performed using intention-to-treat linear mixed-effects models with a group × time interaction term to determine overall differences in effects across time and at each time point. To control for energy intake, all of the foods and nutrient intakes were analysed per MJ of energy intake. The data were analysed using IBM SPSS Statistics (Armonk, NY, USA, Version 23). 

## 3. Results

### 3.1. Outcomes of the MedLey Study

The study population and outcomes of the MedLey study have been previously published [14,15,16,26]. Briefly, following the intervention, there was an increase in the MedDiet adherence score from 7.4 to 10.6 and was significantly higher in the MedDiet group (than HabDiet) at 2 months (between-group difference 2.1, 95% CI 1.3, 2.9, *p* < 0.001) and 4 months (between-group difference 2.6, 95% CI 1.9, 3.3, *p* < 0.001) [14]. Serum β-carotene, lycopene, and erythrocyte monounsaturated fatty acids, measures of compliance to the intervention, were significantly higher in the MedDiet group at 3 and 6 months (*p* < 0.05) [14]. Plasma triglycerides were significantly lower than the HabDiet group at 3 and 6 months (mean difference: −0.15 mmol/L, 95% CI: −0.23, −0.07 mmol/L, *p* < 0.001; and −0.09 mmol/L, 95% CI −0.18, −0.01 mmol/L, *p* = 0.03, respectively). Oxidative stress, as measured by F_2_-isoprostanes, was also lower in the MedDiet group compared with the HabDiet group at 3 and 6 months (mean difference: −103.5 pmol/L, 95% CI −154.2, −52.7 pmol/L, *p* < 0.001; and −65.4 pmol/L, 95% CI: −117.1, −13.7 pmol/L, *p* < 0.001, respectively) [15]. Endothelial function was assessed in 82 participants (*n* = 45 MedDiet, *n* = 33 HabDiet) at baseline and 6 months using FMD. The percentage of FMD was 1.3% higher (95% CI 0.2%, 2.4%, *p* = 0.026) in the MedDiet group at 6 months. Blood pressure was assessed at home, over 6 days, at baseline, 3 and 6 months. Compared with the HabDiet (*n* = 66), the MedDiet (*n* = 70), led to significantly lower systolic blood pressure (P-diet × time interaction = 0.02) [mean −1.3 mmHg (95% CI −2.2, −0.3 mmHg; *p* = 0.008) at 3 months and −1.1 mmHg (95% CI −2.0, −0.1 mmHg; *p* = 0.03) at 6 months] [16]. There were no differences between the groups for total cholesterol and lipoproteins or cognition. 

### 3.2. Follow-Up Population

Of the 137 volunteers who were sent an invitation to participate in the follow-up study, 93% completed the postal questionnaires, and 79% also consented to attend the clinic visit (Figure 1). Of the participants who chose to participate in the follow-up, the average (SD) age was 75 ± 6 years in each group. Overall, 58% and 54% were female in the MedDiet and HabDiet groups, respectively. The MedDiet group had a small, but significantly lower morning (*p* = 0.03), afternoon (*p* = 0.01), evening (*p* = 0.02), and total (*p* = 0.012) diastolic blood pressure than the HabDiet group. There were no other differences in weight, anthropometry, blood lipids, systolic BP, or heart rate. The 20 participants who declined to attend the clinic follow-up had significantly higher mean diastolic BP (*p* = 0.04) and evening diastolic BP (*p* = 0.01) than those who undertook the clinic visit. There were no other differences in anthropometry, blood pressure or blood results.

### 3.3. Mediterranean Diet Adherence

Twelve months after the intervention (referred to as 18 months in this manuscript), the MedDiet group maintained an increased 15-point MedDiet Adherence Score of 7.9 ± 0.3 and an increased 9-point score of 9.2 ± 0.3, which was significantly lower than the 4-month score (*p* < 0.0001) but significantly higher than baseline (*p* < 0.0001) (Figure 2). This pattern was significantly different to the HabDiet group (*p* < 0.0001), who remained unchanged at all time points (Table 1).

### 3.4. Clinical Outcomes

At 18 months, the average home-measured Systolic BP (SBP) increased in the MedDiet group, returning to 121 ± 2 mmHg, which was not significantly different to baseline (Table 2). SBP remained significantly lower at 18 months compared with baseline in the HabDiet group. The MedDiet group had a small but significantly lower overall SBP, and for the morning measure of SBP, than the HabDiet group at 18 months. Diastolic BP (DBP) remained significantly different to baseline at 18 months in the HabDiet group but not in the MedDiet group (Table 2). There was no difference between groups for DBP at 18 months.

Triglycerides were found to be significantly reduced following the MedDiet for 6 months and did not change in the HabDiet group (Table 2). At 18 months, the triglyceride levels in the MedDiet group were no longer significantly lower than baseline values (Table 2). The follow-up study revealed an increase in triglycerides in the HabDiet group at 18 months, from 1.1 ± 0.1 mmol/L at 6 months to 1.2 ± 0.1 mmol/L at 18 months (*p* = 0.032). The MedDiet group had significantly lower triglyceride levels than the HabDiet group at 18 months (diet × visit interaction *p* = 0.02).

HDL cholesterol was not significantly different to baseline at 18 months in the MedDiet group; however, in the HabDiet group, HDL cholesterol was lower at 18 months than at baseline (1.53 ± 0.06 compared to 1.63 ± 0.05 mmol/L, *p* = 0.036). The MedDiet group had a small but significantly higher HDL cholesterol level than the HabDiet group at 18 months (diet × visit interaction *p* = 0.046).

There were no other significant changes observed for blood lipids (total cholesterol, LDL cholesterol, and total cholesterol:HDL), CRP or BMI and body weight across the 18-month period for either group.

The total dietary energy intake for the MedDiet group was reduced following 4 months of dietary intervention (from 8766 ± 264 to 8020 ± 248 kJ/day, *p* = 0.032) and whilst the energy at 18 months was unchanged from 4 months (8040 ± 287 kJ/day, *p* = 1.00), it was no longer significantly different to baseline energy intake (*p* = 0.145) The HabDiet group showed a similar reduction from baseline to 4 months(*p* = 0.036) which was sustained at 18 months (*p* = 0.002) (Table 3).

Protein intake remained unchanged for both dietary groups throughout the study. In the MedDiet group, carbohydrate intake was reduced during the intervention and remained lower than baseline at 18 months. However, the HabDiet exhibited no change in carbohydrates during the study but showed a reduced intake at 18 months compared to baseline (*p* = 0.023) (Table 3). The total dietary fat intake was unchanged in the HabDiet group at 18 months; however, it was significantly higher in the MedDiet group compared to baseline (11.6 ± 0.3 from 10.8 ± 0.2 g/MJ/day, *p* = 0.008). Monounsaturated fat intake increased in the MedDiet group from 4.6 ± 0.1 at baseline to 5.8 ± 0.2 g/MJ/day at 4 months (*p* < 0.0001) and remained elevated at 18 months 5.6 ± 0.2 g/MJ/day (*p* < 0.0001). Saturated fat intake significantly increased at 18 months in the MedDiet group (3.4 ± 0.1 g/MJ/day at 18 months, *p* < 0.0001 compared to 4 months) but was still less than baseline intake (*p* = 0.025) (Table 3). The total polyunsaturated fat intake and alcohol intake remained unchanged for both groups over the course of the intervention. Dietary fibre increased during the MedDiet intervention (2.8 ± 0.1 to 3.0 ± 0.1 g/MJ/day, *p* < 0.0001) but returned to baseline levels by 18 months (*p* = 1.00).

In the MedDiet group, the intakes of extra virgin olive oil (EVOO) and fish at 18 months remained significantly higher than baseline (*p* < 0.0001, *p* = 0.001, respectively) (Table 4, Figure 3). Legume (*p* = 0.006) and vegetable intake (*p* = 0.009) were significantly higher at 18 months compared with the baseline but were significantly lower than at 6 months (*p* < 0.0001, *p* = *0*.032, respectively). Fruit intake at 18 months was higher than the baseline in the MedDiet group but was not significantly different. Similarly, potato intake was lower at 18 months compared with the baseline but not significantly different (*p* = 0.059). There was a significant diet × visit difference for fruit (*p* = 0.048), vegetable (*p* = 0.001) and potato (*p* = 0.001) intake in the MedDiet group. Whilst red meat consumption significantly reduced during the 6-month intervention; its consumption almost returned to baseline intake levels at 18 months. MedDiet increased red wine intake at 3 and 6 months of the intervention (*p* = 0.002) and returned to baseline levels at 18 months. MedDiet discretionary foods decreased significantly during study (*p* < 0.0001), which was maintained at 18 months (*p* = 0.021). Dairy foods, cereals and eggs remained unchanged across all timepoints for the MedDiet. Similarly, all of the food groups remained unchanged across all timepoints for the HabDiet group.

## 4. Discussion

The exploration of the health benefits of the Mediterranean Diet began in the mid-20th century (or the 1950s) by Ancel Keys and has been investigated predominantly in Mediterranean populations. The Mediterranean diet is associated with a lower risk of the incidence of mortality, cardiovascular disease, and dementia and reduces the risk of several other chronic diseases [4,15,16,17,18,29]. Such ailments are modulated by a range of protective bioactive nutrients such as polyphenols, plant sterols, carotenoids, vitamins C and E, fibre, and omega-3-fatty acids [1,4]. The Prevencione con Dieta Mediterranean (PREDIMED) study was the largest RCT in 7668 Spaniards and was conducted over 13 centres across Spain, resulting in a 30% RR in cardiovascular events after a 4.8-year follow-up period [32]. Two of the PREDIMED study arms included a Mediterranean Diet, both of which led to significant improvements in blood pressure, inflammation, oxidative stress, and atherosclerosis when compared to the low-fat control [33]. Following the successful results of the PREDIMED study, many adjunct sub-analyses and derivative studies have emerged, including PREDIMED Plus, PrediDep [34] and PrediBreast, which assessed the role of a Mediterranean dietary pattern for energy restriction, depression, and breast cancer. Because such a healthy diet and lifestyle pattern is conducive to healthy longevity, this dietary pattern offers potential benefits to Western countries that have a high incidence of chronic diseases such as cardiovascular disease, obesity, type 2 diabetes, and depression and has subsequently been implemented into RCTs and longitudinal studies.

Though it has previously been established that Australians can adhere to a Mediterranean Diet and experience health benefits, long-term sustainment tends to waiver in non-Mediterranean populations post-intervention. Although high adherence is observed during intervention periods, sometimes due to complimentary provided food products such as nuts, extra virgin olive oil, and canned tuna, and dietitian visits, once a research study is completed, participants tend to revert to their prior diet and lifestyle habits. Despite the decrease in MEDAS score in the MedDiet group from 4 months 9.6 ± 0.2 to 18 months 7.9 ± 0.3 (*p* < 0.001), the MedLey trial demonstrated that some MedDiet principles could be adhered to up to a year post-intervention. Namely, the MedLey intervention group displayed significantly elevated intakes of total fat, monounsaturated fat, EVOO, vegetables, and fish when compared to baseline and sustained a decreased discretionary food intake at 18 months. However, the intake of foods and food groups that significantly decreased returned to baseline levels or remained unchanged at 18 months may explain the lack of significant clinical outcome results and the subsequent health benefits for participants.

At 18 months, the MedDiet group did not display a significant reduction in blood pressure (SBP and DBP) or blood lipids (HDL cholesterol, total cholesterol, LDL cholesterol, total cholesterol:HDL) compared to baseline, indicating that the participants were no longer following a MedDiet to the extent that they were during the 6-month intervention. Similar results were reported from the AUSMED trial, whereby dairy spreads, poultry, nuts, fish, legumes, wine, fruit, and vegetables slightly decreased compared to baseline and 6 months (end of intervention) [35]. Additionally, the 3-month HELFIMED trial (*n* = 95), a MedDiet intervention (plus fish oil supplementation) targeted at improving the quality of life and mental health for people with depression, demonstrated a capacity to follow some MedDiet principles at 6 months post-intervention (*n* = 85) [20]. The participants’ MEDAS score increased from baseline 4.57 ± 0.24 (low adherence) to 3 months 7.08 ± 0.28 and continued to increase by 6 months 7.44 ± 0.32 (medium adherence).

In the MedLey follow-up, legumes, a key dietary component of the MedDiet, decreased significantly at 18 months compared to 6 months but were still higher than consumption at baseline (*p* < 0.001). Legumes are a good source of plant protein, often used as a meat alternative in main meals. Legumes may be a palatable and easy-to-eat food for older individuals who may be beginning to experience changes in dentition with age. In contrast to the MedDiet, the Western diet is typically high in meat and Australians are said to be one of the highest meat consumers globally [36], consuming almost five servings per week [37]. During the MedLey study, the participants’ red meat intake decreased at 3 and 6 months (*p* < 0.001) but, returned to baseline by 18 months. The 6-month AUSMED Trial similarly saw a decrease in dietary adherence scores at the 12-month follow-up. At the 6-month mark, the MEDAS score for the MedDiet group (*n* = 34) increased to 10.9 ± 1.6 points from the aggregated baseline score of 5.2 ± 2.1 [35]. Of the 34 participants in the intervention group, 27 completed the follow-up at 12 months post-intervention, whereby the MEDAS score had decreased to 9.9 ± 2.1 (medium to high adherence) [35]. The participants with a Mediterranean background reported consuming more legumes compared to those who were not European (*p* = 0.01), and adherence between 6 and 12 months significantly reduced (*p* = 0.02) [35]. These data indicate that Australians may face difficulty in sustaining a low red-meat intake consistent with Mediterranean Diet principles (≤1 serving per week). Further, Australians may feel that legumes are not an appropriate or preferred meat alternative or an acceptable substitute or feel confident in the ways of including legumes in their diet.

The MedLey, AUSMED, HELFIMED, MedPork, and MedDairy trials all demonstrated medium to high adherence during interventions, indicating that a Mediterranean-style diet could be followed post-intervention. However, issues still exist with long-term adoption, which means that individuals may not sustain the health benefits exhibited while following a MedDiet. As we age, we experience changes in our dentition, ability to chew and swallow, and changes to our sense of taste. Interestingly, in a study of 130 Greek individuals of 74 years, a similar age to the MedLey participants, living in open care community centres, the authors found that higher adherence to a MedDiet increased masticatory performance [5]. Dietary support, altering food preparation methods or behaviour modification are strategies to address the challenges that the study participants face during and after a research trial. The tools and strategies learned in behavioural interventions could be useful for the long-term sustainment of diet and lifestyle habits. Such behaviour modification includes motivational interviewing cognitive behavioural therapy (MI-CBT) and could be utilised in dietary intervention trials to increase adherence to MedDiet principles at follow-up. Researchers from London conducted semi-structured focus groups with *n* = 11 participants aged 54 ± 4.0 years old who underwent an 8-week MedDiet and exercise intervention [38]. The participants reported that a MedDiet required a substantial change from their typical routine. The obstacles included increased food costs, travelling commitments, difficulty reading nutrition information panels and labels, increased work, stress and time pressures, difficulty in purchasing food items, and issues with adapting to the MedDiet pattern [38]. Though the average age of the MedLey population was 79 ± 4.9 years old, and the intervention was 6 months as opposed to 8 weeks, it is possible that the participants may have experienced similar barriers to the UK cohort once the MedLey intervention was completed and the participants were no longer receiving intensive, personalised dietetic support. Further mixed methods studies, namely with the inclusion of focus groups, could be employed to determine, what are the barriers to adherence, why adherence to a MedDiet in an Australian context declines over time, and how to overcome this. Additionally, intensive dietetic-led interventions of ≥6 months may be required to promote long-lasting adherence to a Mediterranean diet.

An additional strategy to target behaviour change could include the use of artificially intelligent technology. A 12-week MedDiet and lifestyle intervention in *n* = 31 community dwellers aged 45–75 years old utilised an artificially intelligent virtual heath coach Paola (chatbot) [39]. Over the 12-week intervention, the Mediterranean diet scores increased by an average of 5.7 points and the participants’ anthropometric (body) measures improved [39]. A systematic review of *n* = 15 technology-based nutrition interventions (online and smart application) similarly iterated the potential of technology in delivering targeted Mediterranean diet interventions in the hopes of long-term adoption [40]. However, more research is required, especially in the older population, due to potential issues with technology and smartphone literacy, which may subsequently impact engagement. The use of a co-design with the end users may counteract this issue.

This study has potential limitations. Firstly, the participant-reported outcomes for blood pressure, MDAS score, and FFQ might not be true values, misreported by participants, or subject to bias, though BP was recorded in triplicate and an average was used. Secondly, the budgetary constraints that prevented erythrocyte fatty acid composition, plasma carotenoids, and F_2_-isoprostanes from being analysed could delay a clinically significant finding from being reported. These subsequent analyses would have acted as a supplementary confirmation of if the data collected in the self-reported questionnaires were reflective of the actual participant intake. Further trials are warranted to explore the relationship between age and MEDAS adherence, particularly the role of dental status, which may affect the ability to consume all foods associated with a MedDiet.

## 5. Conclusions

This follow-up study confirmed that a Mediterranean Diet could be adhered to in an Australian population after the cessation of a dietitian-led intervention. Though not all of the principles were sustained at 18 months, at 12 months post-intervention, the participants still achieved a medium adherence diet score, an improvement from their baseline score, and sustained their intake of EVOO, vegetables, and fish, while consuming less discretionary foods. More research is required to determine barriers and enablers to following a MedDiet in the long term and potentially explore more options for meat replacements and substitutions that are widely accepted. Additionally, longer trials (>6 months) in differently-aged populations may be required to encourage the long-term adoption of a MedDiet and ensure that the positive cardiometabolic changes observed during the research trials are maintained.

## Figures and Tables

**Figure 1 nutrients-14-03098-f001:**
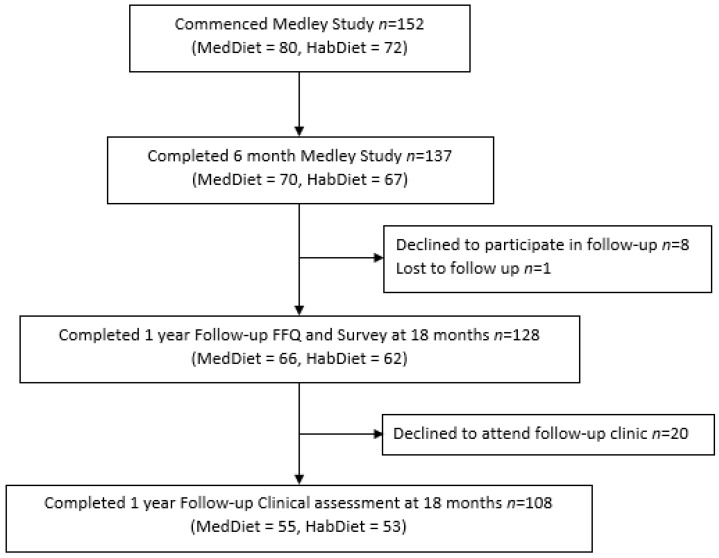
Consort flow diagram for participation in the MedLey study and the subsequent follow-up study.

**Figure 2 nutrients-14-03098-f002:**
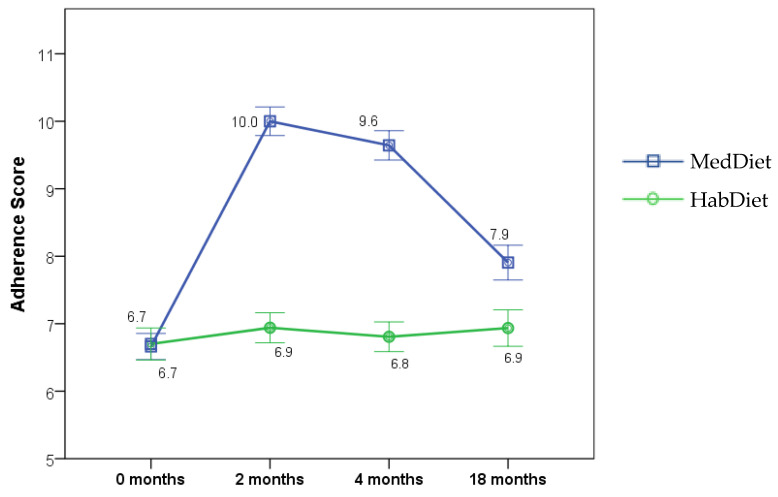
Mediterranean Diet Adherence Score at baseline (0 months) and during 6 months dietary intervention (measured at 2 and 4 months) and 12 months after intervention was ceased (18 months) for MedDiet or HabDiet. Adherence score (mean ± SEM) at each time point by diet group. MedDiet adherence score based on intakes of 15 food groups calculated from food frequency questionnaires, range 0–15, where 15 represents highest possible adherence level. Linear mixed effects model with a group × time interaction term. Analyses were intention to treat. MedDiet: *n* = 77, 70, 70, 66, at 0, 2, 4 and 18-months, respectively; HabDiet: *n* = 70, 67, 67, 62 at 0, 2, 4, and 18 months, respectively. MedDiet, Mediterranean diet; HabDiet, habitual diet.

**Figure 3 nutrients-14-03098-f003:**
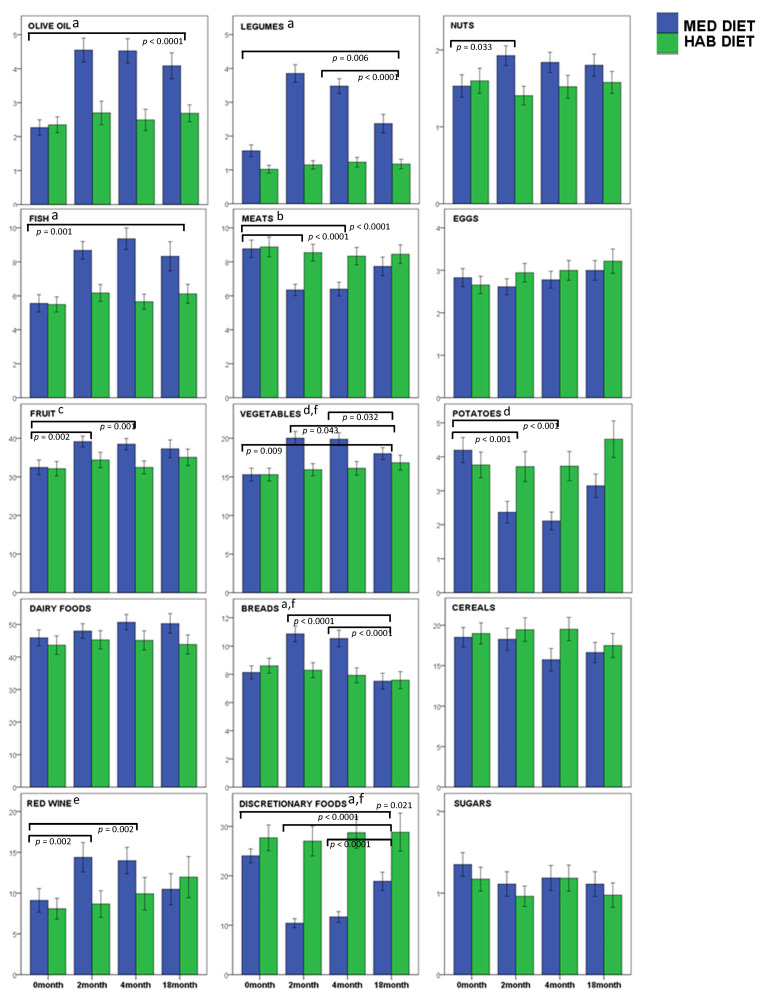
Mean intake of foods for each dietary group over 18 months presented as g/MJ/day. Foods were grouped into 15 categories for the assessment of adherence to a Mediterranean diet (olive oil, legumes, nuts, fish, meats, eggs, fruit, vegetables, potatoes, dairy foods, breads, cereals, red wine, sugars, and discretionary foods). Food intake data were collected by food frequency questionnaires at baseline (0 months), during the 6-months dietary intervention (measured at 2 and 4 months) and 12 months after the intervention ceased (18 months). Data are presented as group mean ± SEM for the Mediterranean diet (MedDiet) intervention and the habitual diet (HabDiet) group. ^a^ diet × visit *p* < 0.0001; ^b^ diet × visit *p* = 0.005; ^c^ diet × visit *p* = 0.048; ^d^ diet × visit *p* = 0.001; ^e^ diet × visit *p* = 0.007; ^f^ 0-months significantly different to 2 and 4 months (*p* < 0.0001).

**Table 1 nutrients-14-03098-t001:** Mediterranean diet adherence measured according to two scores: 15-point score and 9-point literature-based score.

	**MedDiet**										**Diet × Visit** **Interaction**	**Diet**	**Visit**
	**0-Month**	**2-Month**	**4-Month**	**18-Month**	**Change 0–18 Month**			**Change 4–18 Month**					
	**(*n* = 77)**	**(*n* = 70)**	**(*n* = 70)**	**(*n* = 64)**	**Mean Change**	***p*-Value**	**95% CI**	**Mean Change**	***p*-Value**	**95% CI**	***p*-Value**	***p*-Value**	***p*-Value**
15 pt MDAS	6.7 ± 0.2	10.0 ± 0.2	9.6 ± 0.2	7.9 ± 0.3	1.3 ± 0.3	<0.0001	0.6, 2.0	−1.7 ± 0.3	0.000	−2.4, −1.0	<0.0001	<0.0001	<0.0001
9 pt MDAS	7.8 ± 0.3	10.9 ± 0.2	10.3 ± 0.2	9.2 ± 0.3	1.3 ± 0.2	<0.0001	0.7, 2.0	−1.1 ± 0.3	0.000	−1.8, −0.5	<0.0001	<0.0001	<0.0001
	**HabDiet**												
	**0-Month**	**2-Month**	**4-Month**	**18-Month**	**Change 0–18 Month**			**Change 4–18 Month**					
	**(*n* = 70)**	**(*n* = 67)**	**(*n* = 67)**	**(*n* = 62)**	**Mean Change**	***p*-Value**	**95% CI**	**Mean Change**	***p*-Value**	**95% CI**			
15 pt MDAS	6.7 ± 0.2	6.9 ± 0.2	6.8 ± 0.2	6.9 ± 0.3	0.2 ± 0.3	1.000	−0.5, 0.9	0.2 ± 0.3	1.000	−0.5, 0.8			
9 pt MDAS	8.1 ± 0.3	7.9 ± 0.3	7.9 ± 0.3	8.0 ± 0.3	−0.1 ± 0.2	1.000	−0.8, 0.6	0.2 ± 0.3	1.000	−0.5, 0.8			

15-point score (adapted from [28]), 9-point score literature-based score based on [30]. Statistical analysis undertaken by LMEM with a diet × visit interaction comparing MedDiet and HabDiet across 4 time points. Mean values ± SEM. Adherence score based on intakes of food groups calculated from food frequency questionnaire data. MDAS, Mediterranean diet adherence score; MedDiet, Mediterranean diet; HabDiet, habitual diet.

**Table 2 nutrients-14-03098-t002:** Clinical outcomes at baseline (0-months) and during 6 months dietary intervention (measured at 3 and 6 months) and 12 months after intervention was ceased (18 months).

	**MedDiet**										**Diet × Visit** **Interaction**	**Diet**	**Visit**
	**0-Month**	**3-Month**	**6-Month**	**18-Month**	**Change 0–18 Month**			**Change 6–18 Month**					
	**(*n* = 79)**	**(*n* = 73)**	**(*n* = 70)**	**(*n* = 55)**	**Mean Change**	***p*-Value**	**95% CI**	**Mean Change**	***p*-Value**	**95% CI**	***p*-Value**	***p*-Value**	***p*-Value**
SBP (mm/Hg) **	123 ± 1	119 ± 1	117 ± 1	121 ± 2	−2 ± 1	0.219	−5.1, 0.6	4 ± 1	0.000	1.8, 6.4	0.040	0.324	0.000
Morning SBP (mm/Hg)	124 ± 2	120 ± 2	118 ± 2	123 ± 2	−1 ± 1	1.000	−3.9, 2.0	5 ± 1	0.000	2.4, 8.0	0.032	0.360	0.000
Afternoon SBP (mm/Hg)	122 ± 1	117 ± 1	116 ± 1	119 ± 2	−3 ± 1	0.094	−6.4, 0.3	3 ± 1	0.052	−0.01, 5.6	0.093	0.165	0.000
Evening SBP (mm/Hg)	123 ± 2	119 ± 2	117 ± 2	120 ± 2	−3 ± 1	0.136	−6.4, 0.5	4 ± 1	0.003	1.0, 6.8	0.586	0.554	0.000
DBP (mm/Hg) **	70 ± 1	68 ± 1	68 ± 1	68 ± 1	−1 ± 1	0.326	−2.5, 0.4	1 ± 0	0.391	−0.4, 2.2	0.433	0.043	0.000
Morning DBP (mm/Hg)	71 ± 1	70 ± 1	70 ± 1	71 ± 1	0 ± 1	1.000	−1.4, 1.7	2 ± 1	0.023	0.2, 3.2	0.376	0.134	0.000
Afternoon DBP (mm/Hg)	68 ± 1	67 ± 1	66 ± 1	67 ± 1	−1 ± 1	0.542	−3.1, 0.7	1 ± 1	1.000	−1.0, 2.3	0.131	0.011	0.000
Evening DBP (mm/Hg)	69 ± 1	68 ± 1	66 ± 1	67 ± 1	−2 ± 1	0.031	−4.0, −0.1	1 ± 1	1.000	−1.2, 2.4	0.989	0.075	0.000
BMI (kg/m^2^)	26.8 ± 0.4	26.2 ± 0.4	26.2 ± 0.4	26.1 ± 0.5	−0.2 ± 0.1	1.000	−0.5, 0.2	0.1 ± 0.1	1.000	−0.2, 0.3	0.049	0.498	0.000
Weight (kg)	74.0 ± 1.5	72.8 ± 1.5	72.6 ± 1.6	72. 9 ± 1.8	−0.4 ± 0.3	1.000	−1.3, 0.5	0.2 ± 0.3	1.000	−0.6, 0.9	0.019	0.387	0.002
Total cholesterol (mmol/L)	5.14 ± 0.10	4.92 ± 0.10	4.96 ± 0.11	4.84 ± 0.12	−0.2 ± 0.1	0.194	−0.4, 0.4	−0.0 ± 0.1	1.000	−0.3, 0.2	0.256	0.192	0.000
LDL (mmol/L)	2.92 ± 0.08	2.81 ± 0.08	2.83 ± 0.09	2.74 ± 0.10	−0.1 ± 0.1	1.000	−0.3, 0.1	−0.1 ± 0.1	1.000	−0.2, 0.1	0.942	0.105	0.126
HDL cholesterol (mmol/L)	1.63 ± 0.05	1.53 ± 0.05	1.58 ± 0.06	1.59 ± 0.06	1.0 ± 1.0 *	0.767	0.9, 1.0	1.0 ± 1.0 *	1.000	1.0, 1.1	0.046	0.741	0.000
Triglycerides (mmol/L)	1.19 ± 0.05	1.11 ± 0.06	1.04 ± 0.05	1.12 ± 0.07	1.0 ± 1.0 *	1.000	0.9, 1.1	1.1 ± 1.0 *	0.131	1.0, 1.2	0.020	0.508	0.001
Total cholesterol:HDL ratio	3.34 ± 0.10	3.42 ± 0.11	3.35 ± 0.12	3.20 ± 0.11	1.0 ± 1.0 *	1.000	0.9, 1.0	1.0 ± 1.0 *	0.619	0.9, 1.0	0.474	0.553	0.231
CRP (mg/L)	2.09 ± 0.29	1.43 ± 0.20	1.54 ± 0.25	1.44 ± 0.26	0.8 ± 1.1 *	0.170	0.6, 1.1	0.9 ± 1.1 *	1.000	0.7, 1.2	0.297	0.842	0.388
	**HabDiet**												
	**0-Month**	**3-Month**	**6-Month**	**18-Month**	**Change 0–18 Month**			**Change 6–18 Month**					
	**(*n* = 70)**	**(*n* = 68)**	**(*n* = 67)**	**(*n* = 53)**	**Mean Change**	***p*-Value**	**95% CI**	**Mean Change**	***p*-Value**	**95% CI**			
SBP (mm/Hg) **	125 ± 2	122 ± 2	120 ± 2	120 ± 2	−4 ± 1	0.001	−7.1, −1.3	1 ± 1	1.000	−1.5, 3.2			
Morning SBP (mm/Hg)	126 ± 2	123 ± 2	121 ± 2	122 ± 2	−3 ± 1	0.040	−6.2, −0.1	1 ± 1	1.000	−2.0, 3.7			
Afternoon SBP (mm/Hg)	124 ± 2	121 ± 1	119 ± 1	119 ± 2	−5 ± 1	0.001	−8.5, −1.6	0 ± 1	1.000	−2.8, 2.9			
Evening SBP (mm/Hg)	125 ± 2	121 ± 2	118 ± 2	120 ± 2	−5 ± 1	0.004	−8.2, −1.1	2 ± 1	0.591	−1.1, 4.8			
DBP (mm/Hg) **	72 ± 1	71 ± 1	70 ± 1	70 ± 1	−2 ± 1	0.008	−3.3, 0.3	0 ± 0	1.000	−1.2, 1.5			
Morning DBP (mm/Hg)	74 ± 1	73 ± 1	72 ± 1	72 ± 1	−1 ± 1	0.544	−2.7, 0.6	1 ± 1	1.000	0.8, 2.3			
Afternoon DBP (mm/Hg)	71 ± 1	71 ± 1	70 ± 1	68 ± 1	−2 ± 1	0.007	−4.4, −0.5	0 ± 1	1.000	−2.1, 1.2			
Evening DBP (mm/Hg)	71 ± 1	70 ± 1	69 ± 1	69 ± 1	−2 ± 1	0.025	−4.2, −0.2	0 ± 1	1.000	−1.6, 2.2			
BMI (kg/m^2^)	27.1 ± 0.5	26.9 ± 0.5	26.7 ± 0.5	26.9 ± 0.6	−0.1 ± 0.1	1.000	−0.5, 0.2	0.1 ± 0.1	1.000	−0.2, 0.4			
Weight (kg)	75.4 ± 1.6	75.2 ± 1.6	75.0 ± 1.6	75.7 ± 1.9	−0.4 ± 0.3	1.000	−1.3, 0.5	0.1 ± 0.3	1.000	−0.6, 0.9			
Total cholesterol (mmol/L)	5.29 ± 0.10	5.22 ± 0.09	5.14 ± 0.10	5.02 ± 0.11	−0.2 ± 0.1	0.167	−0.4, 0.4	−0.0 ± 0.1	1.000	−0.3, 0.2			
LDL (mmol/L)	3.11 ± 0.09	3.06 ± 0.08	3.06 ± 0.09	2.92 ± 0.08	−0.1 ± 0.1	0.428	−0.3, 0.1	−0.1 ± 0.1	0.684	−0.3, 0.1			
HDL cholesterol (mmol/L)	1.63 ± 0.05	1.60 ± 0.05	1.57 ± 0.04	1.53 ± 0.06	1.0 ± 1.0 *	0.036	0.9, 1.0	1.0 ± 1.0 *	1.000	1.0, 1.0			
Triglycerides (mmol/L)	1.18 ± 0.07	1.21 ± 0.07	1.11 ± 0.06	1.24 ± 0.07	1.1 ± 1.0 *	0.458	1.0, 1.2	1.1 ± 1.0 *	0.032	1.0, 1.2			
Total cholesterol:HDL ratio	3.41 ± 0.10	3.42 ± 0.10	3.40 ± 0.10	3.44 ± 0.11	1.0 ± 1.0 *	1.000	1.0, 1.1	1.0 ± 1.0 *	1.000	1.0, 1.1			
CRP (mg/L)	1.90 ± 0.31	1.85 ± 0.26	1.92 ± 0.30	1.69 ± 0.30	1.0 ± 1.1 *	1.000	0.7, 1.3	1.0 ± 1.1 *	1.000	0.7, 1.4			

Showing statistical analysis by LMEM with a diet × visit interaction comparing MedDiet and HabDiet across 4 time points. Mean values ± SEM. * Non-normally distributed data were analysed as log-transformed data in LMEM, therefore, mean change has been exponentiated and is presented as a ratio of 6 or 18 months versus 0 months (therefore 1.0 means no relative change in geometric mean) for HDL, Triglycerides, Total:HDL cholesterol and CRP. Due to exclusions of outliers the *n* value vary as follows: *n* = −1 for blood pressure measures in HabDiet group, *n* = −3 for LDL (*n* = −2 MedDiet, *n* = −1 HabDiet), *n* = −1 MedDiet for dietary analysis. MedDiet, Mediterranean diet; HabDiet, habitual diet. ** Mean blood pressure measures are the average of 30 daily readings taken in triplicate in morning, afternoon, and night, across 6 days at each time point. Dietary Intake.

**Table 3 nutrients-14-03098-t003:** Daily nutrient intake at baseline (0 months) and during 6-months dietary (measured at 2 and 4 months) intervention and 12 months after intervention ceased (18 months).

	**MedDiet**										**Diet × Visit** **Interaction**	**Diet**	**Visit**
	**0-Month**	**2-Month**	**4-Month**	**18-Month**	**Change 0–18 Month**			**Change 4–18 Month**					
	**(*n* = 77)**	**(*n* = 70)**	**(*n* = 70)**	**(*n* = 64)**	**Mean Change**	***p*-Value**	**95% CI**	**Mean Change**	***p*-Value**	**95% CI**	***p*-Value**	***p*-Value**	***p*-Value**
Energy (kJ/day)	8766 ± 264	8489 ± 267	8020 ± 248	8048 ± 287	−645 ± 283	0.145	−1403, 112	118 ± 247	1.000	−543, 778	0.350	0.696	0.000
Protein (en%)	17.3 ± 0.3	17.1 ± 0.4	17.6 ± 0.4	17.5 ± 0.5									
Fat (en%)	40.0 ± 0.6	41.8 ± 0.9	41.9 ± 1.0	43.0 ± 1.0									
SatFat (en%)	13.3 ± 0.3	10.7 ± 0.2	10.8 ± 0.2	12.5 ± 0.3									
PolyFat (en%)	6.6 ± 0.2	6.8 ± 0.2	6.7 ± 0.2	6.4 ± 0.2									
MonoFat (en%)	17.0 ± 0.5	21.2 ± 0.7	21.3 ± 0.8	20.9 ± 0.8									
CHO (en%)	39.1 ± 0.7	36.8 ± 0.8	36.3 ± 0.8	35.5 ± 0.9									
Alcohol (en%)	4.3 ± 0.5	4.5 ± 0.5	4.6 ± 0.5	4.5 ± 0.7									
Protein (g/MJ/day)	10.2 ± 0.2	10.1 ± 0.2	10.3 ± 0.2	10.3 ± 0.3	0.2 ± 0.2	1.000	−0.4, 0.8	−0.1 ± 0.2	1.000	−0.7, 0.5	0.545	0.745	0.743
Fat (g/MJ/day)	10.8 ± 0.2	11.3 ± 0.3	11.3 ± 0.3	11.6 ± 0.3	0.8 ± 0.2	0.008	0.1, 1.4	0.2 ± 0.2	1.000	−0.3, 0.8	0.320	0.414	0.027
SatFat (g/MJ/day)	3.6 ± 0.1	2.9 ± 0.1	2.9 ± 0.1	3.4 ± 0.1	−0.2 ± 0.1	0.025	−0.5, −0.0	0.4 ± 0.1	0.000	0.3, 0.6	0.000	0.000	0.0000
PolyFat (g/MJ/day)	1.8 ± 0.1	1.8 ± 0.1	1.8 ± 0.1	1.7 ± 0.1	−0.0 ± 0.1	1.000	−0.2, 0.1	−0.1 ± 0.1	0.554	−0.2, 0.1	0.402	0.020	0.338
MonoFat (g/MJ/day)	4.6 ± 0.1	5.7 ± 0.2	5.8 ± 0.2	5.6 ± 0.2	1.0 ± 0.2	0.000	0.6, 1.5	−0.1 ± 0.2	1.000	−0.5, 0.3	0.000	0.005	0.000
CHO (g/MJ/day)	23.0 ± 0.4	21.7 ± 0.5	21.3 ± 0.5	20.9 ± 0.5	−1.9 ± 0.4	0.000	−3.0, −0.9	−0.4 ± 0.4	1.000	−1.3, 0.6	0.207	0.435	0.000
Sugars (g/MJ/day)	11.3 ± 0.3	10.6 ± 0.3	10.8 ± 0.3	10.9 ± 0.4	−0.3 ± 0.3	1.000	−1.0, 0.4	0.2 ± 0.3	1.000	−0.5, 0.9	0.716	0.926	0.049
Alcohol (g/MJ/day)	1.5 ± 0.2	1.6 ± 0.2	1.6 ± 0.2	1.5 ± 0.2	0.1 ± 0.2	1.000	−0.4, 0.5	−0.0 ± 0.1	1.000	−0.4, 0.3	0.504	0.716	0.259
Fibre (g/MJ/day)	2.8 ± 0.1	3.2 ± 0.1	3.0 ± 0.1	2.8 ± 0.1	0.0 ± 0.1	1.000	−0.1, 0.2	−0.2 ± 0.1	0.033	−0.4, −0.0	0.004	0.053	0.001
	**HabDiet**												
	**0-Month**	**2-Month**	**4-Month**	**18-Month**	**Change 0–18 Month**			**Change 4–18 Month**					
	**(*n* = 70)**	**(*n* = 67)**	**(*n* = 67)**	**(*n* = 62)**	**Mean Change**	***p*-Value**	**95% CI**	**Mean Change**	***p*-Value**	**95% CI**			
Energy (kJ/day)	9091 ± 327	8399 ± 301	8321 ± 296	7988 ± 340	−1087 ± 289	0.002	−1861, −313	−311 ± 251	1.000	−983, 362			
Protein (en%)	17.2 ± 0.3	17.4 ± 0.3	17.3 ± 0.3	17.1 ± 0.3									
Fat (en%)	40.6 ± 0.7	41.1 ± 0.9	40.3 ± 0.9	41.2 ± 0.7									
SatFat (en%)	13.8 ± 0.3	13.7 ± 0.3	13.6 ± 0.3	13.8 ± 0.4									
PolyFat (en%)	6.2 ± 0.2	6.3 ± 0.3	6.0 ± 0.2	6.0 ± 0.2									
MonoFat (en%)	17.5 ± 0.5	17.9 ± 0.7	17.7 ± 0.7	18.2 ± 0.6									
CHO (en%)	38.6 ± 0.8	37.6 ± 0.8	38.0 ± 0.8	36.7 ± 0.8									
Alcohol (en%)	4.1 ± 0.5	4.4 ± 0.7	4.9 ± 0.8	5.5 ± 0.9									
Protein (g/MJ/day)	10.1 ± 0.2	10.3 ± 0.2	10.2 ± 0.2	10.1 ± 0.2	−0.1 ± 0.2	1.000	−0.7, 0.5	−0.2 ± 0.2	1.000	−0.8, 0.5			
Fat (g/MJ/day)	11.0 ± 0.2	11.1 ± 0.2	10.9 ± 0.2	11.1 ± 0.2	0.2 ± 0.2	1.000	−0.4, 0.9	0.3 ± 0.2	1.000	−0.3, 0.8			
SatFat (g/MJ/day)	3.7 ± 0.1	3.7 ± 0.1	3.7 ± 0.1	3.7 ± 0.1	0.0 ± 0.1	1.000	−0.2, 0.3	0.1 ± 0.1	1.000	−0.1, 0.3			
PolyFat (g/MJ/day)	1.7 ± 0.1	1.7 ± 0.1	1.6 ± 0.1	1.6 ± 0.1	−0.1 ± 0.1	1.000	−0.2, 0.1	0.0 ± 0.1	1.000	−0.1, 0.2			
MonoFat (g/MJ/day)	4.7 ± 0.1	4.9 ± 0.2	4.8 ± 0.2	4.9 ± 0.2	0.2 ± 0.2	1.000	−0.3, 0.7	0.2 ± 0.2	1.000	−0.3, 0.6			
CHO (g/MJ/day)	22.7 ± 0.4	22.1 ± 0.5	22.4 ± 0.5	21.6 ± 0.5	−1.2 ± 0.4	0.023	−2.3, −0.1	−0.8 ± 0.4	0.169	−1.8, 0.2			
Sugars (g/MJ/day)	11.2 ± 0.4	10.7 ± 0.4	11.0 ± 0.4	10.7 ± 0.4	−0.5 ± 0.3	0.685	−1.2, 0.3	−0.2 ± 0.3	1.000	−0.9, 0.5			
Alcohol (g/MJ/day)	1.4 ± 0.2	1.5 ± 0.2	1.7 ± 0.3	1.9 ± 0.3	0.4 ± 0.2	0.105	−0.0, 0.8	0.2 ± 0.1	1.000	−0.2, 0.5			
Fibre (g/MJ/day)	2.8 ± 0.1	2.8 ± 0.1	2.8 ± 0.1	2.8 ± 0.1	−0.0 ± 0.1	1.000	−0.2, 0.2	−0.0 ± 0.1	1.000	−0.2, 0.2			

Showing statistical analysis by LMEM with a diet × visit interaction comparing MedDiet and HabDiet across 4 time points. Mean values ± SEM. MedDiet, Mediterranean diet; HabDiet, habitual diet.

**Table 4 nutrients-14-03098-t004:** Dietary Intake of grouped foods (g/MJ/day) at baseline (0 months) and during 6 months dietary (measured at 2 and 4 months) intervention and 12 months after intervention ceased (18 months).

	**MedDiet**										**Diet × Visit** **Interaction**	**Diet**	**Visit**
	**0-Month**	**2-Month**	**4-Month**	**18-Month**	**Change 0–18 Month**			**Change 4–18 Month**					
	**(*n* = 77)**	**(*n* = 70)**	**(*n* = 70)**	**(*n* = 64)**	**Mean Change**	***p*-Value**	**95% CI**	**Mean Change**	***p*-Value**	**95% CI**	***p*-Value**	***p*-Value**	***p*-Value**
Dietary Energy (MJ/day)	8.8 ± 0.3	8.5 ± 0.3	8.0 ± 0.2	8.0 ± 0.3	−0.6 ± 0.3	0.148	−1.4, 0.1	0.1 ± 0.2	1.000	−0.5, 0.8	0.350	0.696	0.000
Olive Oil	2.3 ± 0.2	4.5 ± 0.4	4.5 ± 0.4	4.1 ± 0.4	1.7 ± 0.3	0.000	1.0, 2.4	−0.5 ± 0.3	0.438	−1.2, 0.2	0.000	0.001	0.000
Legumes	1.6 ± 0.2	3.9 ± 0.3	3.5 ± 0.2	2.4 ± 0.3	0.8 ± 0.2	0.006	0.2, 1.4	−1.1 ± 0.2	0.000	−1.6, −0.5	0.000	0.000	0.000
Fish	5.6 ± 0.5	8.7 ± 0.5	9.4 ± 0.6	8.3 ± 0.9	2.7 ± 0.7	0.001	0.8, 4.6	−1.1 ± 0.7	0.820	−3.0, 0.8	0.000	0.000	0.000
Meats	8.8 ± 0.5	6.3 ± 0.3	6.4 ± 0.4	7.7 ± 0.5	−1.0 ± 0.6	0.387	−2.5, 0.5	1.3 ± 0.5	0.59	−0.0, 2.5	0.005	0.035	0.000
Fruits	32.5 ± 1.9	39.2 ± 1.4	38.5 ± 1.5	37.3 ± 2.3	4.4 ± 1.8	0.104	−0.5, 9.2	−1.6 ± 1.8	1.000	−6.2, 3.1	0.048	0.149	0.003
Vegetables	15.3 ± 0.8	20.0 ± 0.9	19.9 ± 0.8	18.0 ± 0.8	2.3 ± 0.7	0.009	0.4, 4.3	−1.9 ± 0.7	0.032	−3.7, −0.1	0.001	0.030	0.000
Potatoes	4.2 ± 0.4	2.4 ± 0.3	2.1 ± 0.3	3.1 ± 0.3	−1.2 ± 0.4	0.059	−2.3, 0.0	1.0 ± 0.4	0.074	−0.1, 2.1	0.001	0.015	0.000
Dairy	45.9 ± 2.4	48.0 ± 2.2	50.7 ± 2.3	50.3 ± 3.0	4.6 ± 2.6	0.454	−2.3, 11.5	−0.2 ± 2.7	1.000	−7.5, 7.0	0.672	0.220	0.219
Breads	8.1 ± 0.5	10.9 ± 0.6	10.5 ± 0.6	7.5 ± 0.6	−0.4 ± 0.5	1.000	−1.6, 0.8	−3.0 ± 0.5	0.000	−4.4, −1.6	0.000	0.050	0.000
Cereals	18.5 ± 1.2	18.3 ± 1.4	15.7 ± 1.4	16.6 ± 1.3	−1.3 ± 1.1	1.000	−4.3, 1.6	1.2 ± 1.1	1.000	−1.7, 4.0	0.089	0.397	0.116
Red Wine	9.1 ± 1.5	14.4 ± 1.8	14.0 ± 1.6	10.5 ± 1.9	1.4 ± 1.4	1.000	−2.4, 5.2	−3.0 ± 1.2	0.063	−6.2, 0.1	0.007	0.299	0.007
Eggs	2.8 ± 0.2	2.6 ± 0.2	2.8 ± 0.2	3.0 ± 0.2	0.2 ± 0.2	1.000	−0.4, 0.8	0.2 ± 0.2	1.000	−0.5, 0.8	0.259	0.680	0.117
Sugars	1.4 ± 0.1	1.1 ± 0.2	1.2 ± 0.2	1.1 ± 0.2	−0.1 ± 0.1	0.898	−0.4, 0.1	−0.0 ± 0.1	1.000	−0.3, 0.3	0.597	0.369	0.012
Nuts	1.5 ± 0.1	1.9 ± 0.1	1.8 ± 0.1	1.8 ± 0.1	0.3 ± 0.2	0.508	−0.2, 0.8	−0.0 ± 0.1	1.000	−0.4, 0.3	0.029	0.087	0.718
Other Alcohol	7.4 ± 1.2	2.8 ± 0.7	3.6 ± 0.8	7.7 ± 1.7	−0.2 ± 1.5	1.000	−4.3, 3.9	3.8 ± 1.4	0.036	0.2, 7.5	0.004	0.014	0.008
Discretionary Foods	24.0 ± 1.4	10.4 ± 0.9	11.7 ± 1.1	18.9 ± 1.8	−5.3 ± 1.8	0.021	−10.0, −0.5	6.9 ± 1.6	0.000	2.7, 11.1	0.000	0.000	0.000
	**HabDiet**												
	**0-Month**	**2-Month**	**4-Month**	**18-Month**	**Change 0–18 Month**			**Change 4–18 Month**					
	**(*n* = 70)**	**(*n* = 67)**	**(*n* = 67)**	**(*n* = 62)**	**Mean Change**	***p*-Value**	**95% CI**	**Mean Change**	***p*-Value**	**95% CI**			
Dietary Energy (MJ/day)	9.1 ± 0.3	8.4 ± 0.3	8.3 ± 0.3	8.0 ± 0.3	−1.1 ± 0.3	0.002	−1.9, −0.3	−0.3 ± 0.3	1.000	−1.0, 0.4			
Olive Oil	2.3 ± 0.2	2.7 ± 0.3	2.5 ± 0.3	2.7 ± 0.2	0.4 ± 0.3	0.937	−0.4, 1.2	0.3 ± 0.3	1.000	−0.5, 1.0			
Legumes	1.0 ± 0.1	1.2 ± 0.1	1.2 ± 0.1	1.2 ± 0.1	0.2 ± 0.2	1.000	−0.5, 0.8	−0.0 ± 0.2	1.000	−0.6, 0.6			
Fish	5.5 ± 0.4	6.2 ± 0.5	5.7 ± 0.4	6.1 ± 0.6	0.6 ± 0.7	1.000	−1.3, 2.5	0.4 ± 0.7	1.000	−1.5, 2.4			
Meats	8.9 ± 0.6	8.5 ± 0.5	8.3 ± 0.5	8.4 ± 0.5	−0.5 ± 0.6	1.000	−2.0, 1.1	−0.0 ± 0.5	1.000	−1.3, 1.3			
Fruits	32.1 ± 1.9	34.4 ± 2.0	32.4 ± 1.7	35.1 ± 2.1	3.3 ± 1.8	0.472	−1.7, 8.2	3.2 ± 1.8	0.470	−1.6, 8.0			
Vegetables	15.3 ± 0.8	15.9 ± 0.8	16.1 ± 0.9	16.8 ± 1.0	1.2 ± 0.7	0.532	−0.7, 3.2	0.6 ± 0.7	1.000	−1.2, 2.5			
Potatoes	3.8 ± 0.4	3.7 ± 0.4	3.7 ± 0.4	4.5 ± 0.5	0.7 ± 0.5	0.894	−0.6, 1.9	0.7 ± 0.4	0.545	−0.4, 1.8			
Dairy	43.6 ± 2.8	45.3 ± 2.8	45.1 ± 2.9	43.9 ± 2.9	1.2 ± 2.6	1.000	−5.8, 8.2	−0.8 ± 2.7	1.000	−8.2, 6.6			
Breads	8.6 ± 0.5	8.3 ± 0.5	7.9 ± 0.5	7.6 ± 0.6	−1.1 ± 0.5	0.110	−2.3, 0.1	−0.5 ± 0.5	1.000	−1.9, 1.0			
Cereals	19.0 ± 1.3	19.4 ± 1.4	19.5 ± 1.4	17.5 ± 1.5	−1.8 ± 1.1	0.639	−4.8, 1.2	−2.4 ± 1.1	0.165	−5.4, 0.5			
Red Wine	8.1 ± 1.3	8.7 ± 1.6	9.9 ± 2.0	12.0 ± 2.5	3.2 ± 1.4	0.153	−0.6, 7.1	1.7 ± 1.2	0.927	−1.5, 4.9			
Eggs	2.7 ± 0.2	2.9 ± 0.2	3.0 ± 0.2	3.2 ± 0.3	0.5 ± 0.2	0.111	−0.1, 1.1	0.2 ± 0.3	1.000	−0.5, 0.9			
Sugars	1.2 ± 0.1	1.0 ± 0.1	1.2 ± 0.2	1.0 ± 0.1	−0.2 ± 0.1	0.146	−0.5, 0.0	−0.2 ± 0.1	0.356	−0.5, 0.1			
Nuts	1.6 ± 0.2	1.4 ± 0.1	1.5 ± 0.1	1.6 ± 0.1	−0.0 ± 0.2	1.000	−0.5, 0.5	0.1 ± 0.1	1.000	−0.3, 0.4			
Other Alcohol	10.4 ± 2.3	11.6 ± 2.7	13.0 ± 3.1	14.1 ± 3.6	2.6 ± 1.6	0.570	−1.6, 6.8	0.4 ± 1.4	1.000	−3.3, 4.1			
Discretionary Foods	27.7 ± 2.6	27.0 ± 3.0	28.7 ± 3.2	28.8 ± 3.8	0.8 ± 1.8	1.000	−4.0, 5.6	−0.3 ± 1.6	1.000	−4.5, 4.0			

Showing statistical analysis by LMEM with a diet × visit interaction comparing MedDiet and HabDiet across 4 time points. Mean values ± SEM. Other Alcohol is the sum of all alcohol excluding red wine. Discretionary foods include other alcohol, processed foods high in added sugar, salt and saturated fats. MedDiet, Mediterranean diet; HabDiet, habitual diet.

## Data Availability

The data presented in this study are available on request from the corresponding author.

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
