# Peer review of "Long-Term Adherence to a Mediterranean Diet 1-Year after Completion of the MedLey Study"

_nutrients, 2022, doi:10.3390/nu14153098_

Round 1

Reviewer 1 Report

Although the effort to evaluate the degree of adherence to the Med diet is well recognized, the discussion fails to interpret the findings in a rational manner. In particular:

Methods:

The scoring system of the 9-point literature-based score developed by Sofi et al, as regards the consumption of dairy in this age group for women is not recommended (post-menopause women require the consumption of 2 dairy servings per day).

Statistical methods

Lines 167-169: syntax error . Perhaps the verb is missing

Lines 175-179: Please explain why the non-drinkers were excluded from the analysis 

Follow-up population 

Flow-chart: declined to attend follow-up clinic n=20. Please comment on the characteristics of these participants. Do they differ from the rest who agreed?

Lines 235-237: Please comment on these results in the discussion section

The authors mentioned that the fact that meat consumption and some other food groups returned to the baseline levels could explain the lack of significant clinical outcome results. Do authors imply that some elements of Med Diet are more important than the rest as regards their effects on health? What about the synergistic effects of Med diet components? Or that the increased consumption of legumes cannot counterbalance the increased consumption of meat?

In the present study, the mean age of the participants was 71 years. However, the authors compare the findings with similar studies (30, 31)  in which the characteristics of the participants are significantly different. No comments were made on this issue. 

No justifications are given for the better outcomes of Hab diet participants as compared to Med diet ones.

The physical activity and social life of the participants were not discussed. All these are components of the Med diet lifestyle which could influence the health outcomes.

Not adequate comments are made as regards the results but instead authors provide solutions for intervention studies with the aim to increase compliance. Moreover, the authors extensively discuss potential obstacles/barriers to adherence by providing references to studies conducted with much younger participants although they recognize that the participants' profile is completely different. 

If the elderly cannot significantly improve their dietary habits following the intervention how younger generations could achieve this? This issue was never raised by the authors.

My suggestion is for the authors to revise the discussion section in their attempt to comment on the findings one by one by providing plausible explanations.

Reviewer 2 Report

Dear author,

Thank you to give the opportunity to review your manuscript. Is interesting to analyses the adherence to the Mediterranean diet after a year of intervention.

In general, I recommend to improve the number of references as there is research related to the adherence to the Mediterranean diet that should be included, on the discussion and the references, an example:

Coelho-Júnior HJ, Trichopoulou A, Panza F. Cross-sectional and longitudinal associations between adherence to Mediterranean diet with physical performance and cognitive function in older adults: A systematic review and meta-analysis. Ageing Res Rev. 2021 Sep;70:101395. doi: 10.1016/j.arr.2021.101395. Epub 2021 Jun 19. PMID: 34153553.

Antoniazzi L, Arroyo-Olivares R, Bittencourt MS, Tada MT, Lima I, Jannes CE, Krieger JE, Pereira AC, Quintana-Navarro G, Muñiz-Grijalvo O, Díaz-Díaz JL, Alonso R, Mata P, Santos RD. Adherence to a Mediterranean diet, dyslipidemia and inflammation in familial hypercholesterolemia. Nutr Metab Cardiovasc Dis. 2021 Jun 30;31(7):2014-2022. doi: 10.1016/j.numecd.2021.04.006. Epub 2021 Apr 19. PMID: 34039501.

Bousiou A, Konstantopoulou K, Martimianaki G, Peppa E, Trichopoulou A, Polychronopoulou A, Halazonetis DJ, Schimmel M, Kossioni AE. Oral factors and adherence to Mediterranean diet in an older Greek population. Aging Clin Exp Res. 2021 Dec;33(12):3237-3244. doi: 10.1007/s40520-021-01861-8. Epub 2021 Apr 24. PMID: 33893988.

Cervo MMC, Scott D, Seibel MJ, Cumming RG, Naganathan V, Blyth FM, Le Couteur DG, Handelsman DJ, Ribeiro RV, Waite LM, Hirani V. Adherence to Mediterranean diet and its associations with circulating cytokines, musculoskeletal health and incident falls in community-dwelling older men: The Concord Health and Ageing in Men Project. Clin Nutr. 2021 Dec;40(12):5753-5763. doi: 10.1016/j.clnu.2021.10.010. Epub 2021 Oct 23. PMID: 34763260.

Abstract:

 CVD, is not defined previously and should be before use the acronymous. P should be in lower case an italic.

Introduction:

The Vancouver cites are in superscript and should be font and size as the text.

Methods:

Inclusion and exclusion criteria should be described.

Line 121 to 125: Font text is different.

Line 124: BP is not described before.

Dietary adherence.

Why was not used Medas-14 questionnaire to evaluated the adherence to the Mediterranean Diet.

Statistical analysis

Should be described the statistical analysis applied for the qualitative variables with a parametric a non-parametric distribution.

Results

Should be necessary to described age, sex distribution and education level in the group of MedDiet and Habdiet, to analysed if exist any differences between the groups which may affect the results.

Table 1. Should be interesting include the results of (p value) for the HabDiet and the statistical test used to stablish significative differences (p value). As if it has been done in table 2.

Clinical Outcomes

SBP and DBP have not been described before.

Table 2.  Should be interesting include the results of (p value) for the HabDiet.

Dietary Intake

EVOO has not been described before.

Table 3, 4. Should be interesting include the results of (p value) for the HabDiet and the statistical test used to stablish significative differences (p value).

Figure 3. Should be included the statistical test used to stablish significative differences (p value).

Round 2

Reviewer 2 Report

Dear Authors, 

Changes have been applied correctly. 

Kind regards,